# Influence of the Brewing Temperature on the Taste of Espresso

**DOI:** 10.3390/foods9010036

**Published:** 2020-01-02

**Authors:** Johanna A. Klotz, Gertrud Winkler, Dirk W. Lachenmeier

**Affiliations:** 1Department Life Sciences, University of Applied Sciences Albstadt-Sigmaringen, 72488 Sigmaringen, Germany; johannaklotz@web.de (J.A.K.); winkler@hs-albsig.de (G.W.); 2Chemisches und Veterinäruntersuchungsamt (CVUA) Karlsruhe, 76187 Karlsruhe, Germany

**Keywords:** coffee, espresso, hot beverages, temperature, esophageal cancer, sensory trial

## Abstract

Very hot (>65 °C) beverages such as espresso have been evaluated by the International Agency for Research on Cancer (IARC) as probably carcinogenic to humans. For this reason, research into lowering beverage temperature without compromising its quality or taste is important. For espresso, one obvious possibility consists in lowering the brewing temperature. In two sensory trials using the ISO 4120:2004 triangle test methodology, brewing temperatures of 80 °C vs. 128 °C and 80 °C vs. 93 °C were compared. Most tasters were unable to distinguish between 80 °C and 93 °C. The results of these pilot experiments prove the possibility of decreasing the health hazards of very hot beverages by lower brewing temperatures.

## 1. Introduction

In 1991, coffee was first classified by the International Agency for Research on Cancer (IARC) as “possibly carcinogenic to humans” (group 2B), as there had been a connection to increased risk of bladder cancer [1]. This relationship could not be confirmed in later studies and coffee itself has been reclassified into group 3 “not classifiable” in 2016. In earlier studies, the influence of tobacco smoking had confounded the results of coffee consumption, because both behaviors often occur at the same time [2]. The infusion of mate (*Ilex paraguariensis*) was evaluated as “probably carcinogenic” (group 2A) in 1991 [3]. The significantly increased cancer risk may be based on the fact that mate is typically drunk very hot. Epidemiological studies show that the esophageal cancer risk is increased when mate is consumed very hot, but not when cold [2,4]. Due to this, mate *per se* was included during the 2016 re-evaluation in group 3, similar to coffee *per se*. Animal experiments suggest that a carcinogenic effect occurs at a consumption temperature of 65 °C or higher, which was defined as “very hot” [2,5]. By additionally considering the epidemiological evidence (e.g., [6,7]), the consumption of very hot (>65 °C) beverages independent of type was classified in 2016 as “probably carcinogenic to humans” (group 2A) [2]. Several studies published subsequently to the International Agency for Research on Cancer (IARC) monograph have further strengthened the evidence between the consumption of very hot beverages independent of type and increased esophageal cancer risk [8,9].

In order to avoid the risk of injury in the pharynx due to an excessively high temperature, hot beverages should not be consumed until they have cooled down [10]. In several studies, however, it has been observed that hotter consumption temperatures are often preferred [11]. In a study from southern Germany, the temperature at which coffee is perceived to be too hot was investigated. The consumption temperature of coffee preferred by consumers is 63 °C. The average pain threshold is 67 °C [12]. However, coffee is typically brewed and served at temperatures higher than 65 °C [10,13].

Espresso is a coffee beverage that is usually drunk immediately after brewing and without the addition of milk, which may lower its temperature [14]. Influences on the quality of espresso include the coffee variety (*Coffea arabica* or *C. canephora*) as well as its quality (e.g., defects, origin, etc.), the coffee/water ratio, the water pressure, or grinding grade [15,16,17]. For the extraction of espresso, the water temperature (brewing temperature) had the most significant influence. If the brewing temperature is too high, a higher number of compounds will be extracted into the espresso and its taste will be strongly influenced. Therefore, a maximum brewing temperature of 92 °C has been suggested. At higher brewing temperatures, more bitter and more astringent substances are dissolved into the espresso and its sensory quality is impaired [18]. However, field research detected that temperatures were often set at much higher levels, probably because of unfounded fears about microbiological hazards [13,19,20]. Salamanca et al. confirmed that the bitterness and acidity of espresso was more pronounced at higher brewing temperatures [21]. In a study by Andueza et al., the brewing temperature was also described as the greatest influence on the quality of espresso [22].

With espresso, a lower consumption temperature can be achieved by lowering the brewing temperature. This study will examine whether espresso brewed at 93 °C, for example, differs in taste from espresso brewed at 80 °C.

## 2. Materials and Methods

The basic study design was to investigate a perceptible sensory difference between samples of two products using the forced-choice ISO 4120:2004 sensory analysis methodology “triangle test” [23].

Individuals were given three espresso samples (two temperature low/one temperature high or two temperature high/one temperature low in randomized fashion; levels were either 80 °C vs. 128 °C or 80 °C vs. 93 °C) and asked to make the following decision: Which of the three samples is different? They were additionally asked about their preference regarding the typicity of espresso taste of the deviating sample. The test material for sensory analysis was espresso beans (arabica/canephora mixture, medium dark roast) type Orphea (Maromas group, Tägerwilen, Switzerland). The espresso machine was model ECM Synchronika (Espresso Coffee Machines Manufacture GmbH, Neckargemünd, Germany). Decalcified tap water was used for all trials.

In order to create the same conditions for each espresso extraction according to the Italian Espresso National Institute [24], 7 ± 0.5 g freshly ground coffee powder was weighed directly into the filter holder (type ECM portafilter 1 spout) for each espresso. The grinding degree was adjusted to ensure a percolation time of 25 ± 5 s. The coffee powder was distributed evenly in the filter carrier by vibration. Then, a tamper with a contact pressure of 25 kg was used to press the resulting coffee powder cake. A fine balance placed under the espresso cup was used to ensure the correct quantity of espresso. To start the process, the coffee machine’s brewing lever was turned over. Meanwhile, the balance and stopwatch were observed, and when an espresso quantity of 25 ± 2.5 g was reached, the brewing lever was raised again to stop. If the espresso quantity was below or above the limit, or if the extraction time was outside the specification (25 ± 5 s), a new extraction attempt was started. Particular attention was paid to a consistently uniform preparation method for the sensory trials.

Preliminary tests detected a clearly visible change in color due to the differences in brewing temperature. With a brewing temperature of 80 °C, the espresso was very dark colored with foam on the surface. Espresso at the maximum temperature of 128 °C was rather light brown in color and its consistency as well as the appearance of the foam was also different. For this reason, precautions had to be taken to ensure that during the tastings, the participants did not detect the deviating sample by the existing color deviation. Therefore, a tasting chamber was set up, which prevented light from entering. In addition, two lamps with color-adjustable LED light sources were used. Each color was checked, but only dark blue light, which shone directly into the cups, prevented optical differentiation of the samples. Furthermore, white lids were placed on the espresso cups. The tasters were allowed to only open the lid of one cup at a time, therefore making it impossible to visually compare the samples even when moving them. Before each sample was tasted, the corresponding lid was removed and then replaced.

To ensure that the two identical samples of each triplet actually had identical properties, an espresso extraction with 25 ± 2.5 mL each was divided between the two cups. The deviating sample was also divided, the second sample was used for the next test. Since the coffee machine needs time to heat up or cool down to the desired brewing temperature, it is essential to keep the espresso warm on heating plates until it is tasted, ensuring that all three samples have the same temperature. The test can only be started once the three espresso samples have been equilibrated to the same consumption temperature of approximately 55 °C for a sensory test. Twenty-four people participated in two triangular tests. These included a total of 20 women and four men from different age groups. In the first triangle test, it was tested whether an espresso brewed at 80 °C differed from an espresso brewed at 128 °C. In the second test, the minimum brewing temperature of 80 °C was compared with the setting of 93 °C. 

Power calculations were based on the ISO 4120:2004 [23] protocol and on Schlich [25]. ISO 4120:2004 provides a baseline scenario in which testers were assumed to be able to discriminate with 50% accuracy. To achieve statistical significance at a level of 0.05 for both α-risk (probability of concluding that a perceptible difference exists when one does not) and β-risk (probability of concluding that no perceptible difference exists when one does), at least 23 assessors were needed. For statistical analysis, the results of the espresso discrimination tests were applied to the significance tables of the ISO 4120:2004 based on Meilgaard et al. [26].

## 3. Results

Out of a total of 24 test subjects, 10 individuals identified the deviating sample in both sensory tests. As shown in Table 1, 15 out of 24 people detected a difference between the espresso samples of the first triangular test (80 °C vs. 128 °C). In the second test, the espresso was compared at a brewing temperature of 80 °C with a brewing temperature of 93 °C. Of the 24 test persons, 11 answered this test correctly (Table 1).

## 4. Discussion

According to DIN EN ISO 4120, for a triangular test with a significance level of α = 0.05 and with a number of test persons of *n* = 24, there is a minimum number of correct answers for determining a perceptible difference of 13 persons. It can therefore be concluded that there is a perceptible difference in Test 1 between the espresso sample brewed at 80 °C and the one brewed at 128 °C on the basis of a triangular test.

For the second triangular test, however, since only 11 persons correctly detected a difference in the triangular test, it was not statistically significant. Espresso brewed at 80 °C was not distinguished from espresso brewed at 93 °C by taste.

Limitations of the research design include that the samples were prepared in batches, so that confounding might be introduced by slight differences in preparation and waiting times as well as during sample temperature equilibration until tasting (such as different evaporation of volatiles). These influences were minimized by consistent preparation of the brews in direct sequence and avoidance of evaporation using lids on the cups until tasting.

While the statistical results of specific differences of samples were not included in the research design of triangle tests, it was observed by many tasters during the study that hotter brewed espresso may be described as stronger, more bitter, and more acidic, similar to the study of Salamanca et al. [21]. Our results were comparable to Andueza et al. [22], even though different methodologies were used. In the case of Andueza et al. [22], the espresso samples were extracted at brewing temperatures of 88 °C, 92 °C, 96 °C, and 98 °C. It was found that more solids were detectable in espresso as the temperature increased. The tasting panel found the espresso more bitter and astringent when it was brewed at 96 °C and 98 °C [22]. In addition, in the study of Chapko and Seo, a too hot coffee temperature was described as roasted and burnt [27]. The results of the previous studies correlated with the feedback of the tasting panels in the sensory analysis carried out here.

It is not recommended to extract espresso beyond a brewing temperature of 93 °C. For the samples taken at the setting of the brewing temperature of 128 °C, some negative comments on the sensory attributes were observed, which are burnt, bitter, and strongly acidic. The theoretical background is that the higher the brewing temperature, the more solids and less volatile substances can be dissolved in the espresso, resulting in a negative taste. As a result, more bitter and more astringent flavorings are dominant [18]. It is also interesting to note that the impression can be gained that espresso produced at 80 °C may have been more preferred in the tastings carried out. It might be worthwhile to further test brewing espresso lower than the standard setting of around 90 °C. In this case, the risk of an excessively high consumption temperature can be completely avoided. It is interesting that the Italian Espresso National Institute suggests a temperature of 88 ± 2 °C [24], which is a lower and stricter setting than what Illy and Viani are suggesting (90 ± 5 °C) [18]. However, in practice, at least in many espresso bars in Germany, much higher settings appear to be in common use [13].

## 5. Conclusions

During the sensory examination, it was elucidated that espresso may be brewed less hot for health reasons. The espresso samples that were brewed at lower temperatures could not be distinguished by the tasting panel. For this reason, the coffee machine manufacturers should introduce adjustable brewing temperatures and suggest lower default settings in order to minimize the risk of esophageal cancer and to improve sensory perception. The guideline of the Italian Espresso National Institute, which allows brewing temperatures down to 86 °C, but not over 90 °C, should be more widely implemented [24].

## Figures and Tables

**Table 1 foods-09-00036-t001:** Results of the ISO 4120:2004 sensory analysis using triangle testing for differentiation of espresso prepared using different brewing temperatures.

Brewing Temperature	No. of Assessors	No. of Correct Responses	Significance ^1^	LCI/UCI ^2^
80 °C vs. 128 °C	24	15	yes (α = 0.01)	0.19/0.68
80 °C vs. 93 °C	24	11	no (α = 0.20)	– ^3^

^1^ According to ISO 4120:2004 [23]. For the non-significant trial, the minimum number of correct answers to conclude that a perceptible difference exists (α = 0.05) would have been 13/24. ^2^ Lower and upper 95% confidence intervals (LCI/UCI) for the triangle tests calculated according to ISO 4120:2004 [23]. The limits can be interpreted as percentage of population that can perceive a difference between the samples [26]. ^3^ Not calculated for non-significant trial.

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
