# Peer review of "Influence of the Brewing Temperature on the Taste of Espresso"

_foods, 2020, doi:10.3390/foods9010036_

Round 1

Reviewer 1 Report

Comments
The manuscript deals with the effects of the brewing temperature on the sensory features of espresso coffee.
The Authors have applied the correct process conditions (dose, tamping, extraction time) considering 3 brewing temperature 80, 93 and 128 °C.
The sensorial tests have been well performed.
However, some major revisions should be done to improve the manuscript.

In the Introduction section some more literature should be added concerning the quality of espresso coffee, for example concerning the aspects cited below.

Materials and Methods section
Sensory analyses have been performed in order to assess the taste of espresso, but the Authors should clarify some aspects:

1. the degree of roasting of coffee beans, which has a great influence on the taste of coffee and on its capability to be protected against the modifications due to high temperature (for instance the pH reduction). Is the roasting coffee nearer to the Italian coffee than the American one? The Authors should specify it.
2. the 3 temperatures of brewing should be immediately cited among the process conditions. The comments could become clearer
3. the Authors should explain if the group of extraction has been for 1 or 2 cups. The extraction could be very different as well as the features of obtained espresso
4. the Authors should specify what type of water has been used for brewing 5. the Authors should indicate the granulometry of the powder used for brewing

Furthermore, the Authors have maintained the samples heated until the tasting, but the taste could change during heating! Some volatile compounds could be lost and the pH could decrease, the beverage could concentrate. Were the samples maintained covered during heating?

Even the divided samples could show the same problems.

Finally, the espresso coffee is a typical Italian product, which derives from a specific level of roasting and specific level of grinding of beans as well as from a number of other variables. These aspects greatly affect the taste of coffee beverage and should be standardized as much as possible.

The Authors should make an extra effort to clarify these aspects.

Author Response

The manuscript deals with the effects of the brewing temperature on the sensory features of espresso coffee.

The Authors have applied the correct process conditions (dose, tamping, extraction time) considering 3 brewing temperature 80, 93 and 128 °C.

The sensorial tests have been well performed.

However, some major revisions should be done to improve the manuscript.

In the Introduction section some more literature should be added concerning the quality of espresso coffee, for example concerning the aspects cited below.

Response: Three additional references (#15-17) on espresso quality were included in the introduction as requested.

Materials and Methods section

Sensory analyses have been performed in order to assess the taste of espresso, but the Authors should clarify some aspects:

the degree of roasting of coffee beans, which has a great influence on the taste of coffee and on its capability to be protected against the modifications due to high temperature (for instance the pH reduction). Is the roasting coffee nearer to the Italian coffee than the American one? The Authors should specify it.

Response 1: More details on the beans were included in the methods section as requested.

the 3 temperatures of brewing should be immediately cited among the process conditions. The comments could become clearer

Response 2: The temperature levels were included among the process conditions directly at the beginning as requested.

the Authors should explain if the group of extraction has been for 1 or 2 cups. The extraction could be very different as well as the features of obtained espresso

Response 3: The information was already included. Each extraction was divided between two cups: “To ensure that the two identical samples of each triplet actually have identical properties, an espresso extraction with 25 ± 2.5 ml each is divided between two cups. The deviating sample is also divided, the second sample is used for the next test.” (lines 97-99)

the Authors should specify what type of water has been used for brewing

Response 4: Information on the water was added (standard tap water, decalcified because tap water in Karlsruhe has an extreme degree of hardness).

the Authors should indicate the granulometry of the powder used for brewing

Response 5: Normal grinding degree on a standard coffee mill was chosen, which ensures a percolation time of 25 ± 5 s. This information was added to the text.

Furthermore, the Authors have maintained the samples heated until the tasting, but the taste could change during heating! Some volatile compounds could be lost and the pH could decrease, the beverage could concentrate. Were the samples maintained covered during heating? Even the divided samples could show the same problems.

Response: The samples were directly prepared in sequence, and covered during standing. The tasting was conducted as quickly as possible. Therefore, we believe that there is not a substantial change in this time. Nevertheless, we have included this point as potential limitation in the discussion.

Finally, the espresso coffee is a typical Italian product, which derives from a specific level of roasting and specific level of grinding of beans as well as from a number of other variables. These aspects greatly affect the taste of coffee beverage and should be standardized as much as possible.

Response: Yes, therefore we tried to work as closely to the Italian espresso guidelines as possible (see Odello & Odello, 2006, Ref. 24).

The Authors should make an extra effort to clarify these aspects.

Response: Thank you for all these aspects, which we now all included into the paper.

Reviewer 2 Report

This manuscript presents potentially interesting results, but there are some issues of experimental design, which lead into problems with interpretation and conclusions. 

In this study, a triangle test is the only sensorial trial method that was used. Since the triangle test is designed to detect only an overall difference, not specific differences among three samples, the direction of the difference should NOT be stated as a conclusion of the triangle test.  Authors cannot say if the difference has negative or positive impacts on taste. The specific major issues are as follows.

In introduction, the research questions listed in lines 57-60, “will examine whether a lower brewing temperature of espresso has a negative effect on its taste.~~~” cannot be answered in this study. In method, if one coffee machine was used to make two or three samples (indicated lines 96-98, “Since the coffee machine needs time ~~”), the sample brewings could not be finished at the same time. That means the samples had different waiting time until the samples are tested. If so, the effect of different waiting time on taste/aroma needs to be discussed in the discussion.   In discussion, lines 133-135, “During the sensory analysis carried out in this work, hotter brewed espresso was described as stronger, more bitter~~”, due to the reason stated above, the descriptions from the triangle tests should not be used. Even in lines 142-144, authors’ recommendation was made based on the triangle test panelists’ negative comments on the sensory attributes. As they are not trained panelists and the used tests are not consumer tests (n=24), the data should not be used.       In conclusions, the conclusions were not reasonably drawn. For example, lines 156-157, “The espresso samples that were brewed at lower temperatures are more accepted by the tasting panel”, but the tests that authors used in this study are not the acceptance tests.

Author Response

This manuscript presents potentially interesting results, but there are some issues of experimental design, which lead into problems with interpretation and conclusions.

In this study, a triangle test is the only sensorial trial method that was used. Since the triangle test is designed to detect only an overall difference, not specific differences among three samples, the direction of the difference should NOT be stated as a conclusion of the triangle test.  Authors cannot say if the difference has negative or positive impacts on taste. The specific major issues are as follows.

Response: We partially agree. It is correct that a triangle test per se is not investigating the direction of the difference. However, there are various methodologies available that combine triangle tests with further questions on directions (e.g. see methodology in Dürr P (1986) Sensorische Methoden und ihre statistische Auswertung, in Getränkebeurteilung (Koch J ed), pp 45–81. Eugen Ulmer, Stuttgart, Germany). However, we agree with the reviewer that this has not been the main purpose of our study, and therefore we have deleted the remarks about the secondary decisions from abstract and all appropriate places in the text.

In introduction, the research questions listed in lines 57-60, “will examine whether a lower brewing temperature of espresso has a negative effect on its taste.~~~” cannot be answered in this study.

Response: The lines were deleted.

In method, if one coffee machine was used to make two or three samples (indicated lines 96-98, “Since the coffee machine needs time ~~”), the sample brewings could not be finished at the same time. That means the samples had different waiting time until the samples are tested. If so, the effect of different waiting time on taste/aroma needs to be discussed in the discussion.

Response: The remarks were included as limitation in the discussion, see also response above to similar remarks from reviewer #1.

In discussion, lines 133-135, “During the sensory analysis carried out in this work, hotter brewed espresso was described as stronger, more bitter~~”, due to the reason stated above, the descriptions from the triangle tests should not be used.

Response: We believe that this observation, which was unanimous between the tasters, is worthwhile to mention. However, we agree that due to lack of statistical evaluation in this secondary decision, the arguments need to be put into perspective. Therefore, we have toned down the point and added the mentioned limitations of these findings. It must be mentioned, however, that this finding corroborates previous literature and common knowledge about espresso taste.

Even in lines 142-144, authors’ recommendation was made based on the triangle test panelists’ negative comments on the sensory attributes. As they are not trained panelists and the used tests are not consumer tests (n=24), the data should not be used.

Response: As stated in the previous response, we have toned down the comments in these lines.

In conclusions, the conclusions were not reasonably drawn. For example, lines 156-157, “The espresso samples that were brewed at lower temperatures are more accepted by the tasting panel”, but the tests that authors used in this study are not the acceptance tests.

Response: The wording was changed from “accepted” to “distinguished”, which should be correct due to the methodology.

Round 2

Reviewer 1 Report

The manuscript has been sensitively improved.

The last suggestion concerns the type of extraction. The Authors describe an extraction performed in one step and after divided in two identical cups, but the extraction type expresso could be performed by two different ways (1  or 2 holder filters) and the results in terms of brewing and quality of beverage could be very different.

The Authors should clarify this aspect.

Author Response

The manuscript has been sensitively improved.

The last suggestion concerns the type of extraction. The Authors describe an extraction performed in one step and after divided in two identical cups, but the extraction type expresso could be performed by two different ways (1  or 2 holder filters) and the results in terms of brewing and quality of beverage could be very different.

The Authors should clarify this aspect.

Response: The extraction was performed using a single portion filter. After the extraction, the liquid was divided between the two cups. The used filter type was added to the materials section.

Reviewer 2 Report

I am satisfied with the revision. 

Author Response

I am satisfied with the revision.

Response: Thank you!